# Identifying determinants of under-5 mortality in Bangladesh: A machine learning approach with BDHS 2022 data

**Shayla Naznin**[1]*, **Md Jamal Uddin**[2,3], **Ahmad Kabir**[2]

**1** Department of Statistics, Mawlana Bhashani Science and Technology University, Tangail, Bangladesh, **2** Department of Statistics, Shahjalal University of Science and Technology, Sylhet, Bangladesh, **3** Faculty of Graduate Studies, Daffodil International University, Savar, Dhaka, Bangladesh

* shayla.naznin@gmail.com

## Abstract

### Background

Under-5 mortality in Bangladesh remains a critical indicator of public health and socio-economic development. Traditional methods often struggle to capture the complex, non-linear relationships influencing under-5 mortality. This study leverages advanced machine learning models to more accurately predict under-5 mortality and its key determinants. By enhancing prediction accuracy, the study aims to provide actionable insights for improving child survival outcomes in Bangladesh.

### Methods

Multiple machine learning (ML) algorithms were applied to data from the 2022 Bangladesh Demographic Health Survey, including Random Forest, Decision Tree, K-Nearest Neighbors, Logistic Regression, Support Vector Machine, XGBoost, LightGBM and Neural Networks. Feature selection was performed using the Boruta algorithm and model performance was evaluated by comparing accuracy, precision, recall, F1 score, MCC, Cohen's Kappa and AUROC.

### Results

The Random Forest (RF) model emerged as the most effective predictive model for under-5 mortality in Bangladesh, surpassing other models in various performance metrics. The RF model delivered impressive results, achieving 98.75% Accuracy, 98.61% Recall, 98.88% Precision, 98.74% F1 Score, 97.5% MCC, 97.5% Cohen's Kappa and an AUROC of 99.79%. These metrics highlight its exceptional predictive accuracy and robustness. Key factors influencing under-5 mortality identified by the model included the number of household members, wealth index, parents' education

**Data availability statement:** The data used in this study are from the Bangladesh Demographic and Health Survey (BDHS) 2022, which is publicly available through the Demographic and Health Surveys (DHS) Program. Due to ethical and legal restrictions imposed by the DHS Program, the authors cannot share the dataset directly. However, researchers can request access by registering on the DHS Program website: https://dhspro-gram.com/data/available-datasets.cfm.

**Funding:** The author(s) received no specific funding for this work.

**Competing interests:** The authors declare that they have no competing interests.

(both father's and mother's), the number of antenatal care (ANC) visits, birth order and the father's occupation.

## Conclusions

The Random Forest model excelled in predicting under-5 mortality in Bangladesh identifying key predictors such as household size, wealth, parental education, ANC visits, birth order and father's occupation. These findings underscore the efficacy of machine learning in predicting under-5 mortality and identifying critical determinants these also provide a data-driven foundation for policymakers to design targeted interventions, such as improving access to maternal healthcare, promoting parental education and addressing socio-economic inequalities, ultimately contributing to enhanced child survival outcomes in Bangladesh.

## Introduction

A crucial measure of a society's development and overall well-being is the rate of mortality among the children under five years old. Under-five mortality (U5M), expressed per 1,000 live births, refers to the likelihood of a child dying before reaching the age of five. In developing countries like Bangladesh, U5M remains a critical public health issue, despite notable improvements in recent decades. Bangladesh has made substantial progress, with the U5M rate decreasing from 126 per 1,000 live births in 1994–31 in 2022 [1], driven by successful public health initiatives such as widespread immunization, disease control programs, and enhanced maternal and child healthcare services [2,3]. In particular, maternal education has played a pivotal role in reducing child mortality, as higher levels of maternal education are strongly linked to better health outcomes for children [4].

Despite these advances, achieving the target of reducing U5M to 25 per 1,000 live births by 2030, as outlined by the Sustainable Development Goals (SDGs) [5], remains a significant challenge. Entrenched socio-economic disparities continue to influence child survival outcomes [6]. Research has identified several key determinants of U5M in Bangladesh, including maternal education, father's education, wealth index, birth order, maternal age, birth spacing, access to healthcare services and geographical location [7]. For example, children born to mothers with lower educational attainment or in rural areas are more vulnerable to mortality due to reduced access to healthcare and resources [8]. Socio-economic status, sanitation, and breastfeeding practices are other crucial factors shaping child survival [9].

Traditional studies have played a crucial role in identifying risk factors for under-five mortality. However, they often struggle to account for the complex, non-linear interactions between variables that influence child survival [2,10,11]. In contrast, machine learning (ML) has emerged as a powerful analytical tool capable of uncovering hidden patterns in large datasets, handling complex interactions, and improving predictive accuracy. Unlike traditional regression-based

models, ML techniques can incorporate a wide range of variables and automatically detect intricate relationships that may otherwise go unnoticed [12–17]. Studies in other developing countries, such as India and Ethiopia, have demonstrated the potential of ML techniques to identify key predictors of child mortality, such as maternal education [18], household resources and access to clean water [12], providing valuable insights for policymakers and healthcare professionals.

To address this gap, this study leverages data from the 2022 Bangladesh Demographic and Health Survey (BDHS) [1] and employs machine learning (ML) algorithms to analyze under-five mortality in the country. Specifically, this research aims to: (i) identify the key determinants of under-five mortality in Bangladesh using ML techniques. (ii) enhance the accuracy of mortality predictions by leveraging advanced ML models.

By applying Machine Learning methodologies, this study seeks to provide data-driven insights that can inform targeted public health interventions and contribute to reducing child mortality. The findings will support evidence-based policymaking in line with SDG targets, ultimately improving child survival outcomes in Bangladesh.

## Methods and materials

### Data

The data for this research were drawn from the Bangladesh Demographic and Health Survey (BDHS) 2022 the latest installment in a series of nationally representative surveys conducted every three to four years since 1993. The BDHS surveys, being cross-sectional in nature, utilized a two-stage stratified random sampling method that ensured representation from all administrative divisions throughout Bangladesh. The BDHS surveys, with nearly identical questionnaires across rounds, enable consistent comparisons of demographic and health indicators, such as maternal education, over time. Comprehensive details regarding the BDHS sampling techniques and methodology have been published in earlier publications. No new data collection was carried out by the authors.

Since the BDHS survey collects detailed birth histories, mothers were first asked whether they had ever given birth. Women who had never given birth were excluded from the study. If a mother had multiple children, the most recent birth within the five years preceding the survey was selected to ensure consistency in exposure to maternal and household conditions. The prenatal care data were collected for this most recent child.

### Sample size and handling missing data

For this study, we accessed the publicly available BDHS 2022 dataset [1], which is nationally representative and covers the entire country we extracted data for 8,839 weighted female respondents. Individuals identified as temporary residents (de jure population) and cases with missing information were excluded to ensure data quality. After removing incomplete cases, a total of 4,913 respondents remained for machine learning analysis.

### Target variable

The primary variable of interest in this study was under-5 mortality, which refers to the childs deaths prior to their fifth birthday. To capture this, mothers of childbearing age were asked, "Is your child alive?" Their responses were coded as "no" (0) if the child had died before the age of five and "yes" (1) if the child was still alive. In other words, the variable was recorded as 1 for child deaths occurring under the age of five and 0 if the child survived beyond that age. A total of 8839 (weighted) children were included in the study, among whom 268 children had died before their fifth birthday. Women without children were excluded from the analysis.

### Independent variables

Based on the previous study we consider under-5 mortality independent variables are follows:

| Variables | Categories/Coding | Description |
|---|---|---|
| Division | 0=Barisal, 1=Chattogram, 2=Dhaka, 3=Khulna, 4=Rajshahi, 5=Mymensingh, 6=Rangpur, 7=Sylhet | Administrative regions of Bangladesh |
| Place of residence | 0=Urban, 1=Rural | Geographic location where the household resides |
| Mother's educational level[1] | 0=No, 1=education, 2=Primary, 3=Secondary, 4=Higher | Highest level of education completed by the mother |
| Wealth index[2] | 0=Poorest, 1=Poorer, 2=Middle, 3=Richer, 4=Richest | Composite economic status based on asset ownership |
| Currently breastfeeding | 0=No, 1=Yes | Whether the child is being breastfed at the time of the survey |
| Father's education level[1] | 0=No, 1=education, 2=Primary, 3=Secondary, 4=Higher | Highest level of education completed by the father |
| Religion | 1=Muslim, 2=Non-Muslim | Religious affiliation of the mother |
| Sources of Drinking Water | 1=Other than Tube well or Borehole, 2=Tube well or Borehole | Type of primary drinking water source |
| Mother's occupation | 1=Worker/Labor, 2=Professional Worker, 3=Business & Others | Type of employment of the mother |
| Father's occupation | 1=Agriculture, 2=Worker/Labor, 3=Professional Worker, 4=Business & Others | Type of employment of the father |
| Type of Toilet Facilities | 1=Toilet With Flush, 2=Ventilated Improved Pit latrine (VIP), 3=Pit Latrine, 4=Hanging toilet and Other | Sanitation facility available in the household |
| Type of Cooking | 1=Natural Gas/ Kerosene, 2=Wood, 3=Agricultural Crop, 4=Animal Dung and Others | Primary fuel used for cooking |
| Main floor material | 1=Earth/Sand, 2=Cement and Others | Primary material of the household floor |
| Main wall material | 1=Cane/Palm/Trunks/Dirt/Bamboo with Mud, 2=Tin, 3=Cement and Others | Primary material of the household walls |
| Main roof material | 1=Tin/Thatch/Palm Leaf, 2=Cement and Others | Primary material of the household roof |
| **Mother's** age at first birth | 1=Below 18 years old, 2=Above 18 years old | Age of the mother at the birth of her first child |
| Marriage to first birth interval | 1=Below 12 months, 2=12–36 months, 3=37–60 months, 4=60 months and above | Time interval between marriage and first childbirth |
| Age at first Marriage | 1=<15, 2=15-19, 3=19-24, 4=24+ | Age at which the mother first got married |
| Birth order number | 1=One, 2=Two, 3=Three or more | Birth order of the child |
| Duration of breastfeeding | 1=Below 1 months, 2=1–12 months, 3=13–24 months, 4=24 months and above | Total duration of breastfeeding for the child |
| Place of delivery | 1=Home, 2=Other than Home | Location where the child was born |
| Number of antenatal visits during pregnancy | 1=No, 2=1–3, 3=4–6, 4=6+ | Number of antenatal checkups received by the mother |
| Sex of child | 1=Male, 2=Female | Biological sex of the child |
| Body Mass Index | 1=Thin(<18.5), 2=Normal (18.5–24.99), 3=Overweight (25–29.99), 4=Obese(>=30) | Nutritional status of the mother based on BMI |
| Access to Media | 0=No, 1=Yes | Whether the mother has access to TV, radio, or newspapers |

## Statistical analysis

**Machine learning model implementation.** This study applies various machine learning classification models to analyze the risk factors associated with under-five mortality. The models include Random Forest, Decision Tree, Logistic Regression, KNN, SVM, XGBoost, LightGBM and Neural Networks. To implement these machine learning models, we split the entire dataset into training and testing subsets, allocating 70% of the data for training and 30% for testing. This division was facilitated by the `train_test_split` function from the `sklearn.model_selection` module in Python, ensuring a randomization process that is crucial for unbiased model evaluation. By employing stratified sampling based on the target variable, we preserved the distribution of classes in both the training and testing datasets and by setting a fixed random seed (random_state), we ensured reproducibility of our results across multiple iterations.

 

**Addressing class imbalance.** Addressing class imbalance was a critical component of the data preprocessing phase. The initial dataset revealed a significant skew, with 96% of the samples representing live children and only 4% representing cases of under-five mortality. To rectify this imbalance, we utilized the Synthetic Minority Over-sampling Technique (SMOTE), which allowed us to achieve a more balanced distribution of classes, resulting in an equitable 50:50 ratio.

**Hyperparameter optimization.** To ensure optimal model performance, **GridSearchCV** was used for hyperparameter tuning. The table below presents the final hyperparameter values for each model:

| Model | Key Hyperparameters |
|---|---|
| **Random Forest** | random_state = 42 |
| **Decision Tree** | random_state = 42 |
| **KNN** | Default parameters |
| **Logistic Regression** | random_state = 42 |
| **SVM** | kernel = 'linear', random_state = 42, probability = True |
| **XGBoost** | random_state = 42, use_label_encoder = False, eval_metric = 'logloss' |
| **LightGBM** | random_state = 42 |
| **Neural Network** | random_state = 42, max_iter = 500 |

**Evaluation metrics and external validation.** To evaluate model performance, we employed several metrics including accuracy, sensitivity, specificity, precision, F1-score, Matthews correlation coefficient (MCC), Cohen's Kappa and the area under the ROC curve (AUC). For more reliable and robust model evaluation, we implemented k-fold cross-validation.

**Machine learning vs. traditional statistical methods.** Additionally, to understand the comparative performance of machine learning and traditional statistical methods, we conducted bivariate analysis along with chi-square tests. This combination enabled us to assess the significance of variables related to under-five mortality.

Multivariate analysis was not performed as machine learning models inherently capture complex interactions between variables without requiring explicit multivariate regression. Traditional methods, such as logistic regression, assume linear relationships, whereas models like Random Forest and XGBoost handle non-linearity and multicollinearity effectively. Additionally, feature selection using the Boruta algorithm ensured the inclusion of only the most relevant predictors, making separate multivariate analysis unnecessary.

**Feature selection.** Furthermore, we employed the Boruta algorithm to identify the key features contributing to under-five mortality, ultimately revealing eight significant variables. This rigorous approach enhances the reliability and generalizability of our findings, providing valuable insights into the factors influencing under-five mortality.

**Software and implementation.** We incorporated all twelve significant variables into the application of eight machine learning models, including Random Forest, Decision Tree, KNN, Logistic Regression, SVM, XGBoost, LightGBM and Neural Networks, utilizing Python software and its version 3.0. We employed the Boruta algorithm through the Boruta package in the R programming language to select the most relevant features, as it is specifically designed for feature selection in a way that takes into account the interactions between features. Using different software for specific tasks allowed us to leverage each platform's strengths: Python for machine learning and R for statistical analysis and feature selection with Boruta. This combination enhances the rigor of our analysis, providing comprehensive insights into the factors influencing under-five mortality.

## Machine learning models and feature selection techniques

**Decision tree (DT).** The Decision Tree (DT) algorithm utilizes a hierarchical, tree-like structure to classify data by recursively splitting features based on decision rules [19]. Each internal node represents a decision point, while leaf nodes denote final classifications. DT is advantageous for its interpretability and ability to handle both categorical and continuous

variables. However, it is prone to overfitting, particularly when the tree grows excessively deep. In this study, DT helps identify key decision points influencing fertility outcomes.

**Random forest (RF).** Random Forest (RF) is an ensemble learning method that constructs multiple decision trees and aggregates their outputs to enhance predictive accuracy [20]. By averaging results from multiple trees, RF mitigates overfitting and improves generalization. It is particularly effective for handling high-dimensional data and complex interactions between variables, making it well-suited for analyzing fertility determinants.

**Support vector machine (SVM).** Support Vector Machine (SVM) is a robust classification algorithm that constructs an optimal hyperplane to maximize the margin between different classes [21]. It is particularly effective in high-dimensional spaces and is well-suited for datasets with non-linear decision boundaries through the use of kernel functions. In fertility prediction, SVM ensures high accuracy by efficiently capturing complex relationships between socio-economic and demographic factors.

**Logistic regression (LR).** Logistic Regression (LR) is a widely used probabilistic model for binary classification problems, estimating the probability of an outcome based on input variables [22]. Due to its simplicity and interpretability, LR is particularly useful for examining the influence of individual predictors on fertility outcomes. It provides clear insights into the significance and direction of each determinant's effect.

**K-nearest neighbors (KNN).** K-Nearest Neighbors (KNN) is a non-parametric, instance-based learning algorithm that classifies a data point based on the majority class among its k-nearest neighbors [23]. While simple and effective for small datasets, KNN is sensitive to feature scaling, outliers and high-dimensional data, which can impact classification performance. Despite these limitations, KNN offers a flexible approach for fertility predictions.

**Extreme gradient boosting (XGBoost).** XGBoost is an optimized gradient boosting framework designed to enhance predictive performance while minimizing overfitting [24]. By leveraging regularization techniques and parallel processing, XGBoost achieves high accuracy and efficiency, particularly for structured data. Its robustness and computational efficiency make it a strong candidate for fertility determinant analysis.

**Light gradient boosting machine (LightGBM).** LightGBM is a highly efficient gradient boosting framework that uses a novel histogram-based learning technique to improve speed and memory efficiency [25]. Unlike traditional boosting methods, LightGBM builds trees in a leaf-wise rather than level-wise manner, allowing for better performance on large datasets. Its ability to handle categorical variables directly enhances model interpretability and efficiency in fertility prediction.

**Neural networks (NN).** Neural Networks (NN), inspired by biological neural structures, consist of interconnected layers that learn patterns in data [26]. The NN model implemented in this study is a Multi-Layer Perceptron (MLP) with input, hidden and output layers. The hidden layers utilize the ReLU activation function and the model is trained using backpropagation. NNs excel at capturing complex, non-linear relationships between fertility determinants, but they require substantial computational resources and careful tuning to avoid overfitting.

**Model evaluation metrics.** The confusion matrix provides a comprehensive evaluation of classification performance by comparing actual versus predicted outcomes [27].

We selected the following key metrics to provide a more comprehensive evaluation:

**Accuracy-**While commonly used, accuracy alone is inadequate in imbalanced datasets. However, it provides a baseline measure of overall model correctness when class distributions are considered.

**Sensitivity (Recall)-** Given that under-five mortality is a rare event, identifying actual cases is crucial. Sensitivity (or recall) measures the proportion of correctly identified under-five mortality cases (true positives) among all actual cases. A higher recall indicates that fewer cases of under-five mortality are missed.

**Specificity-** This metric measures the model's ability to correctly classify cases where the child survives (true negatives). While improving recall is essential, maintaining a balance with specificity ensures that the model does not misclassify too many healthy children as deceased.

**Precision-** Precision evaluates the proportion of correctly predicted under-five mortality cases out of all predicted cases. A high precision ensures that false positives are minimized, which is critical when designing policy interventions to target high-risk populations.

**F1-score-** This is the harmonic mean of precision and recall, providing a balanced metric when dealing with class imbalances. A high F1-score ensures that both false positives and false negatives are minimized, making it a more reliable measure than accuracy alone.

**Matthews Correlation Coefficient (MCC)-** MCC provides a single-value summary of model performance that accounts for all four confusion matrix components (true positives, true negatives, false positives, false negatives). It is particularly useful for imbalanced datasets, as it remains balanced even when class distributions are uneven.

**Cohen's Kappa-** This metric measures agreement between predicted and actual classifications while adjusting for random chance. Given that imbalanced datasets can inflate apparent performance; Cohen's Kappa offers a more adjusted assessment.

### Receiver operating characteristic (ROC) curve and area under the curve (AUC)

The ROC curve visually represents the trade-off between true positive and false positive rates across various classification thresholds [28]. The Area Under the Curve (AUC) quantifies model performance, with higher AUC values indicating superior classification ability. This metric is essential for comparing machine learning models in fertility predictions.

### Feature selection: Boruta algorithm

The Boruta algorithm, a feature selection method based on Random Forest, was employed to identify the most relevant predictors while eliminating irrelevant ones [29]. Unlike Lasso or Recursive Feature Elimination (RFE), Boruta evaluates feature importance by comparing actual variables against randomly permuted "shadow features." This process ensures that no potentially significant determinant of fertility is overlooked, making it particularly effective for high-dimensional datasets.

### Ethics approval and consent to participate

Data for this study was accessed with authorization from the Measure DHS program following legal registration. The analysis used the 2022 Bangladesh Demographic and Health Survey (BDHS) dataset, which is publicly available through the Measure DHS website (https://dhsprogram.com/data/available-datasets.cfm).

## Results

### Descriptive outcomes of the background characteristics

The frequency distributions of mothers and under-five mortality, alongside their associated chi-square statistics and p-values, are detailed in Table 1. The analysis of various characteristics associated with under-five mortality reveals significant insights. In Barishal, 96.84% of children did not experience under-five mortality, while in Chattogram, the figure was 96.94%, with both regions showing low percentages of mortality at 3.16% and 3.06%, respectively. Dhaka's statistics indicated a slight increase in mortality rates, showing 2.53% for those who experienced under-five mortality. The data also highlighted that urban residents had a marginally lower mortality rate (2.93%) compared to rural residents (3.04%). Mother's education played a crucial role, with higher mortality observed among mothers with primary education (3.45%).

**Table 1. Descriptive Statistics: Frequencies, Percentages and Chi-square Test Results with p-values.**

| Variables | Total | Under-five Mortality | | χ2 | p-value |
|---|---|---|---|---|---|
| | | No | Yes | | |
| **Division** | n(%) | n(%) | n(%) | | |
| Barishal | 570(6.45) | 552(96.84) | 18(3.16) | | |
| Chattogram | 1925(21.78) | 1866(96.94) | 59(3.06) | | |
| Dhaka | 2216(25.07) | 2160(97.47) | 56(2.53) | 11.835 | 0.106 |
| Khulna | 888(10.05) | 869(97.86) | 19(2.14) | | |
| Mymensingh | 750(8.48) | 727(96.93) | 23(3.07) | | |
| Rajshahi | 907(10.26) | 881(97.13) | 26(2.87) | | |
| Rangpur | 963(10.89) | 925(96.05) | 38(3.95) | | |
| Sylhet | 621(7.02) | 593(95.49) | 28(4.51) | | |
| **Type of place of residence** | | | | | |
| Urban | 2386(26.99) | 2316(97.07) | 70(2.93) | | |
| Rural | 6453(73.01) | 6257(96.96) | 196(3.04) | 0.064 | 0.800 |
| **Mother's educational level** | | | | | |
| No education | 549(6.21) | 531(96.72) | 18(3.28) | | |
| Primary | 2059(23.29) | 1988(96.55) | 71(3.45) | 7.428 | 0.059 |
| Secondary | 4743(53.66) | 4595(96.88) | 148(3.12) | | |
| Higher | 1488(16.83) | 1459(98.05) | 29(1.95) | | |
| **Number of household members** | | | | | |
| 1-3 Member | 1153(13.05) | 1089(94.45) | 64(5.55) | | |
| 4 Member | 1987(22.48) | 1924(96.83) | 63(3.17) | 33.944 | <0.001 |
| 5 Member | 1979(22.39) | 1938(97.93) | 41(2.07) | | |
| 6 Member | 1344(15.21) | 1305(97.1) | 39(2.9) | | |
| Over 6 Member | 2375(26.87) | 2316(97.52) | 59(2.48) | | |
| **Wealth index** | | | | | |
| Poorest | 1849(20.91) | 1768(95.62) | 81(4.38) | | |
| Poorer | 1838(20.79) | 1772(96.41) | 66(3.59) | 24.362 | <0.001 |
| Middle | 1821(20.6) | 1770(97.2) | 51(2.8) | | |
| Richer | 1702(19.25) | 1668(98) | 34(2) | | |
| Richest | 1631(18.45) | 1596(97.85) | 35(2.15) | | |
| **Children Ever Born** | | | | | |
| 1-3 Children | 7853(88.85) | 7640(97.29) | 213(2.71) | | |
| 4-6 Children | 924(10.45) | 875(94.7) | 49(5.3) | 21.656 | <0.001 |
| Over 6 Children | 61(0.69) | 57(93.44) | 4(6.56) | | |
| **Mother's age at first birth** | | | | | |
| < 18 years | 2977(33.68) | 2880(96.74) | 97(3.26) | 0.953 | 0.329 |
| ≥ 18 years | 5862(66.32) | 5693(97.12) | 169(2.88) | | |
| **Marriage to first birth interval** | | | | | |
| < 12 months | 1670(19.23) | 1622(97.13) | 48(2.87) | | |
| 12 - 36 months | 5077(58.46) | 4934(97.18) | 143(2.82) | 14.327 | 0.002 |
| 37 - 60 months | 1365(15.72) | 1328(97.29) | 37(2.71) | | |
| 60 + months | 572(6.59) | 540(94.41) | 32(5.59) | | |
| **Age at first Marriage** | | | | | |
| Below 18 | 5650(63.92) | 5465(96.73) | 185(3.27) | 3.766 | 0.052 |
| 18 and above | 3189(36.08) | 3108(97.46) | 81(2.54) | | |

*(Continued)*

**Table 1.** (Continued)

| Variables | Total | Under-five Mortality | | χ2 | p-value |
|---|---|---|---|---|---|
| | | No | Yes | | |
| **Father's education level** | | | | | |
| No education | 1387(15.94) | 1330(95.89) | 57(4.11) | | |
| Primary | 2675(30.74) | 2584(96.6) | 91(3.4) | 21.334 | <0.001 |
| Secondary | 3017(34.67) | 2927(97.02) | 90(2.98) | | |
| Higher | 1623(18.65) | 1600(98.58) | 23(1.42) | | |
| **Birth order number** | | | | | |
| One | 3372(38.15) | 3276(97.15) | 96(2.85) | | |
| Two | 3011(34.07) | 2934(97.44) | 77(2.56) | 7.016 | 0.030 |
| Three or more | 2455(27.78) | 2363(96.25) | 92(3.75) | | |
| **Sex of child** | | | | | |
| Male | 4527(51.21) | 4378(96.71) | 149(3.29) | 2.326 | 0.127 |
| Female | 4313(48.79) | 4195(97.26) | 118(2.74) | | |
| **Number of antenatal visits during pregnancy** | | | | | |
| No | 408(7.96) | 388(95.1) | 20(4.9) | | |
| 1-3 | 2689(52.44) | 2640(98.18) | 49(1.82) | 19.736 | <0.001 |
| 4-6 | 1592(31.05) | 1558(97.86) | 34(2.14) | | |
| 6+ | 439(8.56) | 435(99.09) | 4(0.91) | | |
| **Place of delivery** | | | | | |
| Home | 1952(36.11) | 1880(96.31) | 72(3.69) | | |
| Other than Home | 3454(63.89) | 3380(97.86) | 74(2.14) | 11.345 | 0.001 |
| **Religion** | | | | | |
| Muslim | 8147(92.17) | 7897(96.93) | 250(3.07) | | |
| Non-Muslim | 692(7.83) | 676(97.69) | 16(2.31) | 1.25 | 0.263 |
| **Father's occupation** | | | | | |
| Agriculture | 1721(19.75) | 1657(96.28) | 64(3.72) | | |
| Laborer/ Worker | 4521(51.89) | 4379(96.86) | 142(3.14) | 8.88 | 0.031 |
| Professional Worker | 601(6.9) | 590(98.17) | 11(1.83) | | |
| Business/ Others | 1869(21.45) | 1825(97.65) | 44(2.35) | | |
| **Mother's occupation** | | | | | |
| House Wife | 6159(69.69) | 5988(97.22) | 171(2.78) | | |
| Laborer/ Worker | 1711(19.36) | 1649(96.38) | 62(3.62) | 4.085 | 0.252 |
| Professional Worker | 672(7.6) | 648(96.43) | 24(3.57) | | |
| Business/ Others | 296(3.35) | 287(96.96) | 9(3.04) | | |
| **Access to media** | | | | | |
| No | 3843(43.47) | 3713(96.62) | 130(3.38) | 3.048 | 0.081 |
| Yes | 4997(56.53) | 4860(97.26) | 137(2.74) | | |

Note:

[1]The BDHS follows Bangladesh's education system, where primary education typically includes 1–5 years, secondary includes 6–10 years and higher education includes 11+years.

[2]Derived based on household assets, housing conditions and access to basic services. Followed standard BDHS measure.

## Features selection

Fig 1 illustrates the feature importance rankings derived from the Boruta algorithm, highlighting the significant variables associated with under-five mortality prediction. This analysis revealed 12 critical variables, including factors such as the number of household members, birth order number, wealth index, place of delivery, father's education, father's occupation, mother's education, place of residence, number of ANC visit, total child ever born, age at first marriage, mother's occupation have a notable influence on the under-five mortality in Bangladesh. These identified variables are essential for the subsequent evaluation of machine learning models, enhancing our understanding of their predictive power regarding under-five mortality.

## Model performance comparison for predicting under-5 mortality

Several machine learning algorithms were employed to develop a predictive model for under-5 mortality in Bangladesh, including Random Forest, Decision Tree, KNN, Logistic Regression, SVM, XGBoost, LightGBM, and Neural Networks. Each model was trained using 70% of the data, with the remaining 30% used for testing. The performance of several machine learning models (Table 2) was evaluated using various metrics, including accuracy, precision, recall, F1-score, AUC (Area Under the Curve) (Fig 2), MCC (Matthews Correlation Coefficient), and Cohen's Kappa. The goal was to assess their ability to classify data into two classes, with the Random Forest, XGBoost, and LightGBM models emerging as the top performers. Below is a detailed interpretation of the results.

Random Forest achieved the highest accuracy (0.9875), indicating that nearly 99% of the predictions were correct. The precision, recall, and F1-scores for both classes were also high (all approximately 0.99), suggesting that the model performed well in identifying both classes with minimal misclassification. The AUC score of 0.9979 reflects the model's excellent ability to distinguish between the two classes. Similarly, MCC (0.9750) and Cohen's Kappa (0.9750) demonstrate that Random Forest produced highly reliable predictions, with very strong agreement between the predicted and actual classes. Overall, Random Forest demonstrated exceptional performance, making it the most reliable model in this evaluation.

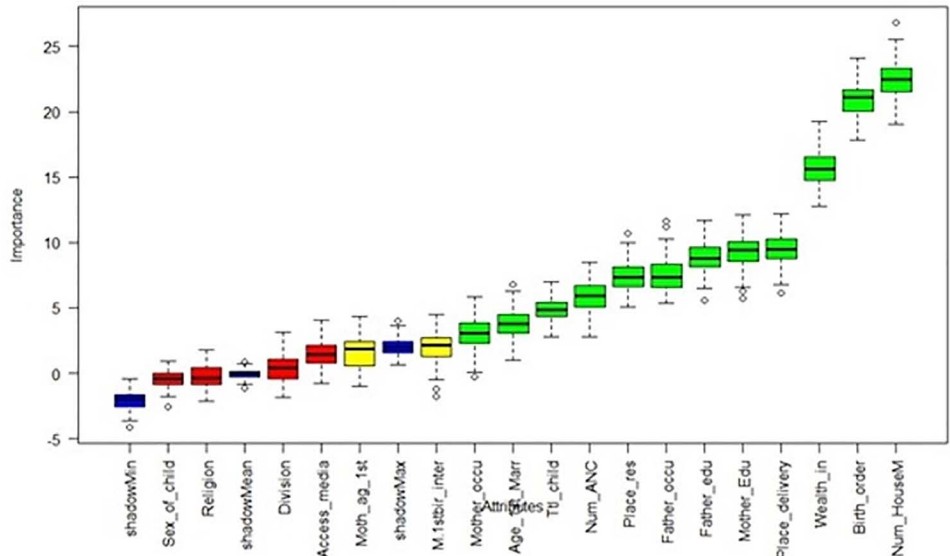

**Fig 1. Feature selection using Boruta algorithm.**

**Table 2. The predictive models performance for under-5 mortality with 95% confidence intervals.**

| Model | Random Forest | Decision Tree | KNN | Logistic Regression |
|---|---|---|---|---|
| Accuracy | 0.9875(±0.0038) | 0.9629(±0.0071) | 0.9140(±0.0099) | 0.6964(±0.0168) |
| AUC | 0.9979(±0.0015) | 0.9648(±0.0068) | 0.9687(±0.0066) | 0.7797(±0.0170) |
| MCC | 0.975(±0.0076) | 0.9265(±0.0141) | 0.8383(±0.0172) | 0.3931(±0.0334) |
| Cohen Kappa | 0.9750(±0.0076) | 0.9258(±0.0142) | 0.8282(±0.0194) | 0.3928(±0.0335) |
| Precision | 0.9888(±0.0054) | 0.9457(±0.0114) | 0.8578(±0.0160) | 0.6882(±0.0241) |
| Recall | 0.9861(±0.0061) | 0.9819(±0.0069) | 0.9916(±0.0049) | 0.7131(±0.0230) |
| F1-score | 0.9874(±0.0039) | 0.9634(±0.0071) | 0.9199(±0.0101) | 0.7004(±0.0189) |
| Model | SVM | XGBoost | LightGBM | Neural Network |
| Accuracy | 0.6932(±0.0175) | 0.9795(±0.0054) | 0.9809(±0.0051) | 0.9307(±0.0090) |
| AUC | 0.7791(±0.0169) | 0.9980(±0.0009) | 0.9977(±0.0010) | 0.9741(±0.0050) |
| MCC | 0.3880(±0.0351) | 0.9591(±0.0107) | 0.9619(±0.0101) | 0.8646(±0.0170) |
| Cohen Kappa | 0.3867(±0.0350) | 0.9591(±0.0108) | 0.9619(±0.0100) | 0.8614(±0.0179) |
| Precision | 0.6774(±0.0240) | 0.9811(±0.0070) | 0.9785(±0.0074) | 0.8962(±0.0148) |
| Recall | 0.7326(±0.0232) | 0.9777(±0.0079) | 0.9833(±0.0067) | 0.9735(±0.0085) |
| F1-score | 0.7039(±0.0188) | 0.9794(±0.0056) | 0.9809(±0.0051) | 0.9332(±0.0091) |

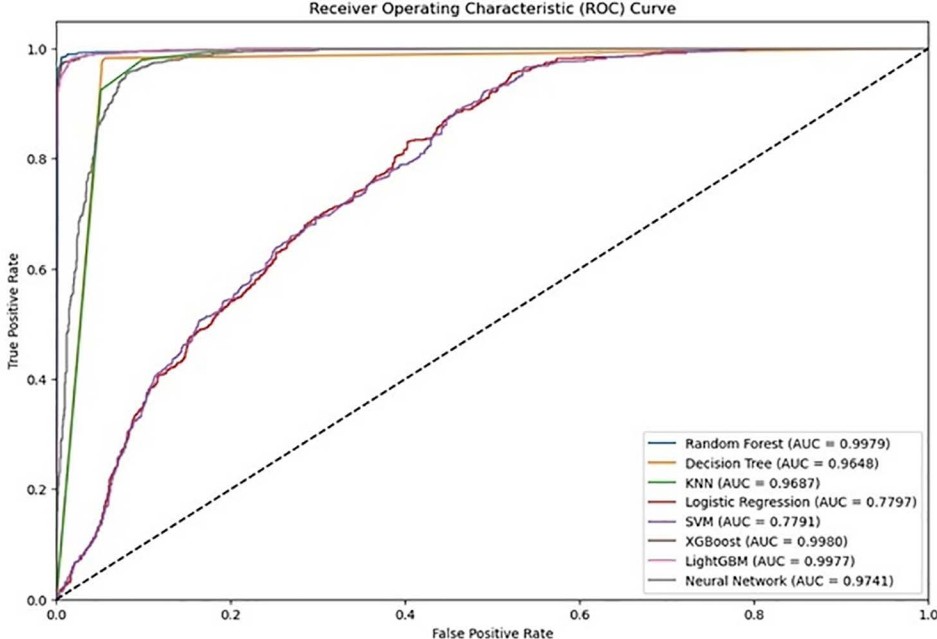

**Fig 2. ROC Curves for Under-5 Mortality Prediction Models.**

XGBoost also performed extremely well, with an accuracy of 0.9795, slightly lower than Random Forest but still impressive. Its precision, recall, and F1-scores for both classes were equally high (around 0.98), indicating that it performed well in predicting both classes. The AUC score of 0.9980 was the highest among all models, reflecting an almost perfect ability to distinguish between the two classes. The MCC (0.9591) and Cohen's Kappa (0.9591) also confirmed strong model

performance. Though slightly below Random Forest, XGBoost remained a top performer, capable of handling the classification task with high precision.

LightGBM was another strong model, with an accuracy of 0.9809. Its precision, recall, and F1-scores were similarly high for both classes (around 0.98). The AUC of 0.9977 and MCC of 0.9619 demonstrated that LightGBM also performed excellently, with very few misclassifications. The Cohen Kappa of 0.9619 showed that LightGBM was in strong agreement with the actual labels. LightGBM performed on par with XGBoost, making both models highly suitable for this classification task.

Decision Tree, while performing well, had a slightly lower accuracy of 0.9629. The recall (0.98), precision (0.95) and F1-score remained high (0.96), showing that the model still managed to maintain good performance. The AUC score of 0.9648 and MCC of 0.9265, though lower than those of Random Forest, XGBoost, and LightGBM, suggest that Decision Tree was a reasonably effective model but not as robust as the top performers.

K-Nearest Neighbors (KNN) demonstrated a decent performance with an accuracy of 0.9140. The precision (0.86), recall (0.99), AUC (0.9687), MCC (0.8383) and Cohen Kappa (0.8282), signaling that KNN is less reliable overall compared to Random Forest, XGBoost, and LightGBM.

Logistic Regression and Support Vector Machine (SVM) were the weakest models in this evaluation. Logistic Regression achieved an accuracy of 0.6964, with precision and recall values hovering around 0.70 for both classes. This indicates that the model was only able to correctly classify around 70% of the data, with frequent misclassifications. The AUC score of 0.7797, along with an MCC of 0.3931 and Cohen Kappa of 0.3928, suggests that Logistic Regression did not perform well on this dataset, making it an unreliable choice for this task. Similarly, SVM achieved an accuracy of 0.6932, with slightly lower recall and precision values compared to Logistic Regression. Its AUC score of 0.7791, MCC of 0.3880, and Cohen Kappa of 0.3867 confirm that SVM was also inadequate for this classification task.

Lastly, the Neural Network model performed moderately well with an accuracy of 0.9307. Its precision (0.90) and recall (0.97), AUC (0.9741), MCC (0.8646) suggest that while the Neural Network outperformed Logistic Regression, SVM, and KNN, it was not as reliable as Random Forest, XGBoost, or LightGBM.

**K-fold cross validation**

In our evaluation of various machine learning models using K-fold cross-validation with fold values of 5, 10, 15, 20, 25, and 30, Random Forest, LightGBM, and XGBoost emerged as the top performers, demonstrating consistent and high accuracy across all folds in the Table 3. Random Forest achieved scores between 0.9764 and 0.9770, while LightGBM and XGBoost exhibited similarly stable results, ranging from 0.9752 to 0.9772. These models displayed strong generalization capabilities and minimal sensitivity to changes in the number of folds. Support Vector Machines (SVM) and Neural

**Table 3. K-Fold Cross Validation Accuracy of Different Models.**

| Model | 5 Folds | 10 Folds | 15 Folds | 20 Folds | 25 Folds | 30 Folds |
|---|---|---|---|---|---|---|
| Random Forest | 0.9770 | 0.9764 | 0.9766 | 0.9768 | 0.9768 | 0.9770 |
| Gradient Boosting | 0.9760 | 0.9758 | 0.9754 | 0.9762 | 0.9760 | 0.9752 |
| Decision Tree | 0.9548 | 0.956 | 0.9534 | 0.9558 | 0.9534 | 0.9546 |
| KNeighbors | 0.8351 | 0.8296 | 0.8221 | 0.8274 | 0.8248 | 0.8233 |
| Logistic Regression | 0.6674 | 0.6593 | 0.6505 | 0.6591 | 0.6564 | 0.6483 |
| SVM | 0.9623 | 0.9601 | 0.9595 | 0.9609 | 0.9609 | 0.9621 |
| XGBoost | 0.9760 | 0.9752 | 0.9756 | 0.9758 | 0.9754 | 0.9762 |
| LightGBM | 0.9762 | 0.9762 | 0.9770 | 0.9762 | 0.9772 | 0.9770 |
| Neural Network | 0.9625 | 0.9611 | 0.9638 | 0.9621 | 0.9605 | 0.9621 |

Networks also performed reliably, with only slight fluctuations in accuracy, making them solid choices for this task. Conversely, Decision Trees showed more variability, with performance dipping at higher fold values, ranging from 0.9534 to 0.9560. K-Nearest Neighbors (KNN) and Logistic Regression performed the worst, with KNN showing a consistent decline in accuracy as the number of folds increased, and Logistic Regression demonstrating a gradual drop from 0.6674 to 0.6483. These findings suggest that while Random Forest, LightGBM, and XGBoost are the most effective models for this task, KNN and Logistic Regression are less suited for optimal performance due to their weaker generalization and declining accuracy with higher fold values.

### Features importance

Fig 3 presents the feature importance rankings from a Random Forest classifier, identifying key factors influencing under-five mortality. Household size is the most important factor, likely due to resource allocation and healthcare access, followed by wealth index, which highlights the impact of socio-economic status. Father's education and antenatal care visits also play crucial roles, emphasizing the importance of parental education and maternal healthcare. Birth order, maternal education, and father's occupation further contribute, reflecting socio-economic and cultural influences.

### Discussion

The primary objective of this study was to predict the determinants of under-5 mortality in Bangladesh using machine learning methods, with a focus on the data 2022 Bangladesh Demographic and Health Survey (BDHS). This study explores the various socio-economic, demographic, and health-related factors that influence under-5 mortality, offering significant insights into how these factors interact in the context of Bangladesh. The results, presented in Table 1, highlight key associations that provide critical evidence for targeted public health interventions.

The analysis revealed significant associations between under-5 mortality and various household and socio-economic characteristics. Notably, a significant relationship was found between the number of household members and child mortality, with those in smaller households (1–3 members) exhibiting the highest mortality rate (5.55%, $\chi^2 = 33.944$,

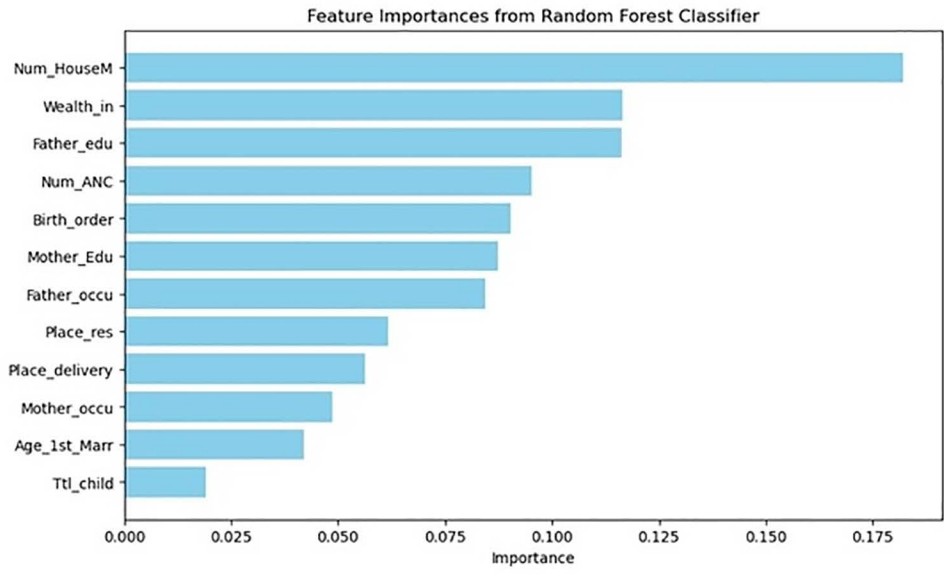

**Fig 3. Visualization of Important Features Using the Random Forest Classifier.**

p < 0.001). Similarly, wealth status proved to be a significant predictor, with the poorest households exhibiting the highest mortality rate (4.38%, $\chi^2 = 24.362$, p < 0.001). Furthermore, children born into families with four to six siblings experienced higher mortality (5.3%, p < 0.001), suggesting a strain on family resources as a potential contributing factor. Conversely, maternal age at first birth and the interval from marriage to first birth showed no significant associations with under-5 mortality, though age at first marriage showed a near-significant effect (p = 0.052) for those marrying before the age of 18 [30]. Education levels of both parents emerged as significant, particularly paternal education (p < 0.001), emphasizing the importance of parental education in mitigating child mortality. Furthermore, place of delivery was a crucial factor; home births showed a higher mortality rate (3.69%) compared to facility-based deliveries (2.14%). Religion showed no significant impact on mortality outcomes, with minimal differences observed between Muslims and non-Muslims.

These findings align with previous studies conducted in South Asian contexts, which have consistently highlighted the role of parental education, household wealth, and healthcare access in reducing child mortality rates [12,15,16,31]. Studies from Nepal and India have similarly found that children from wealthier households and those with educated parents experience significantly lower mortality rates [32,33]. This reinforces the broader evidence base that socio-economic factors are critical determinants of child survival.

In terms of predictive modeling, several machine learning models were employed to predict under-5 mortality, including Logistic Regression (LR), Support Vector Machine (SVM), K-Nearest Neighbors (KNN), Random Forest (RF), Decision Tree (DT), Gradient Boosting (GB), Adaptive Boosting (AdaBoost), Neural Networks (NN), Extreme Gradient Boosting (XGBoost), and Light Gradient Boosting Machine (LightGBM). Among these, ensemble methods such as Random Forest and Gradient Boosting demonstrated superior performance in predicting under-5 mortality, as evidenced by higher precision, recall, and F1 scores when compared to traditional models like Logistic Regression and Decision Trees [24,34]. These results are consistent with other studies applying machine learning to health-related datasets, where Random Forest and Gradient Boosting have demonstrated robustness in handling complex interactions and non-linearity [13,14]. Similar studies in maternal and child health have found boosting techniques to be particularly effective in imbalanced datasets, as they enhance predictive accuracy through iterative learning processes [15].

Particularly, Random Forest and Gradient Boosting showed remarkable robustness in handling complex interactions between variables and in capturing non-linear relationships, which are crucial in health-related datasets. XGBoost and LightGBM also performed well, benefiting from their advanced boosting techniques that efficiently manage imbalanced data, such as our dataset, where the number of under-5 deaths was significantly lower compared to the survival cases. However, models like KNN and SVM struggled with the imbalanced nature of the data, resulting in lower performance metrics, as expected from algorithms that do not inherently handle class imbalances well.

The machine learning models identified several key predictors that significantly influence under-5 mortality in Bangladesh. Household characteristics, such as the number of household members and wealth index, emerged as important determinants [35–38]. Larger households were associated with higher under-5 mortality, potentially due to resource constraints and challenges in providing adequate care for multiple children. Socio-economic status, as measured by the wealth index [39], showed a strong inverse relationship with child mortality, with children from wealthier households experiencing lower mortality risks. These findings are consistent with previous research in Bangladesh and other low-income countries, where socio-economic inequalities are strongly linked to disparities in child survival rates [7,39]–41].

Parental factors, particularly father's education and mother's education, also played a significant role in determining under-5 mortality [9,32,42–44]. Higher levels of maternal and paternal education were associated with lower child mortality, likely reflecting better health knowledge, access to healthcare, and family planning. This reinforces the critical role of education, particularly for women, in improving child health outcomes. Additionally, father's occupation was identified as a contributing factor, with more stable and higher-status occupations linked to reduced child mortality risk which is support to the previous study [40].

Access to healthcare services, measured by the number of antenatal care (ANC) visits, also emerged as a crucial determinant [6,16,35,45,46]. Mothers who received more ANC visits were less likely to experience child mortality, highlighting the importance of ensuring comprehensive maternal healthcare access throughout pregnancy. ANC visits provide opportunities for early detection and management of health risks, contributing to better maternal and child health outcomes. Similar findings have been reported in previous studies in Bangladesh and Sub-Saharan Africa, where increased ANC visits were strongly correlated with improved birth outcomes and reduced child mortality [11].

Finally, birth order emerged as a key demographic factor, with higher birth order being associated with increased under-5 mortality. This aligns with previous research [8,9,33,47,48] suggesting that resource allocation within families may become strained as the number of children increases, leading to poorer health outcomes for subsequent children.

The findings from this study have several policy implications. First, the strong association between parental education, particularly maternal education, and under-5 mortality underscores the need for continued investments in education as a long-term strategy for reducing child mortality. Educating parents not only enhances their decision-making capacity regarding health and family planning but also improves child health outcomes. Second, improving access to maternal and child healthcare services should remain a priority. Expanding antenatal care coverage, promoting skilled birth attendance, and ensuring affordable healthcare, particularly in rural and underserved regions, could significantly reduce under-5 mortality. Third, targeted interventions addressing socio-economic disparities and high fertility rates may be necessary. Previous policy interventions, such as cash transfer programs and community-based maternal health initiatives, have been effective in other countries and could be further explored in Bangladesh.

This study demonstrates the potential of machine learning in predicting determinants of under-5 mortality in Bangladesh. By leveraging the 2022 BDHS data, we were able to identify key socio-economic, demographic, and healthcare-related factors that significantly contribute to child mortality. Ensemble learning methods like Random Forest and Gradient Boosting were particularly effective in capturing the complex interplay of factors, providing robust predictions. These results reinforce the growing body of literature advocating for machine learning applications in public health [12,13,15], particularly in child mortality prediction. The insights from this study underscore the importance of continued efforts to improve parental education, access to healthcare, and family planning services to further reduce under-5 mortality in Bangladesh. Future research should explore longitudinal data and more sophisticated machine learning techniques to refine these predictions and strengthen the evidence base for policymaking.

## Strengths and limitations

This research has several strengths, including the application of advanced machine learning models, which allowed for better prediction of under-5 mortality using a comprehensive dataset from the 2022 BDHS. The use of oversampling techniques to handle imbalanced data improved model reliability, and the policy-relevant insights provided valuable guidance for targeting health interventions. However, limitations include the cross-sectional nature of the data, which limits causal inference, and challenges with model interpretability. Despite efforts to mitigate overfitting, the results may lack generalizability to other settings, and some models struggled with the data imbalance, reducing their effectiveness. These factors suggest areas for future research improvement.

## Conclusion

In summary this study reveals the effectiveness of machine learning models in predicting the determinants of under-five mortality in Bangladesh, utilizing data from the 2022 Bangladesh Demographic and Health Survey (BDHS). By employing a range of machine learning models, including ensemble methods like Random Forest and Gradient Boosting, we were able to identify key socio-economic, demographic, and healthcare-related factors that significantly influence child mortality.

In conclusion, we have identified several key factors influencing under-5 mortality, including the number of household members, wealth index, parental education (both father's and mother's), the number of antenatal care (ANC) visits, birth order and the father's occupation.

The findings of this study offer several important policy implications. Investments in parental education, particularly for women, enhancing access to maternal and child healthcare and addressing socio-economic disparities should be prioritized to further reduce under-5 mortality rates. Moreover, targeted family planning interventions can help mitigate the risks associated with high fertility and larger households.

Looking ahead, future research should focus on incorporating longitudinal data to provide more robust predictions over time and explore more sophisticated machine learning techniques to refine the models. These insights can support policy-makers and healthcare practitioners in designing evidence-based strategies to improve child health outcomes and reduce under-5 mortality in Bangladesh.

## Author contributions

**Conceptualization:** Shayla Naznin, Md Jamal Uddin.

**Data curation:** Shayla Naznin.

**Formal analysis:** Shayla Naznin.

**Methodology:** Shayla Naznin.

**Supervision:** Md Jamal Uddin, Ahmad Kabir.

**Validation:** Shayla Naznin.

**Visualization:** Shayla Naznin.

**Writing – original draft:** Shayla Naznin.

**Writing – review & editing:** Shayla Naznin.

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
