## [Decision Letter · Decision Letter 0]

PONE-D-24-48508Predicting Determinants of Under-5 Mortality in Bangladesh Using Machine Learning Algorithm: Insights from BDHS 2022PLOS ONE

Dear Dr. Naznin,

Thank you for submitting your manuscript to PLOS ONE. After careful consideration, we feel that it has merit but does not fully meet PLOS ONE’s publication criteria as it currently stands. Therefore, we invite you to submit a revised version of the manuscript that addresses the points raised during the review process. It is recommended to explain how your selected model will be useful for policy making.

We look forward to receiving your revised manuscript.

Kind regards,

Md. Moyazzem Hossain, PhD

Academic Editor

PLOS ONE

3. In the online submission form, you indicated that [Availability of data and materials

The manuscript includes all relevant data. However, the primary data supporting the

findings of this manuscript will be made available upon request. The data used in this

study is derived from the publicly accessible BDHS 2022 datasets,

available through the Measure DHS website (https://dhsprogram.com/data/availabledatasets.cfm).].

Reviewers' comments:

Reviewer's Responses to Questions

**Comments to the Author**

1. Is the manuscript technically sound, and do the data support the conclusions?

Reviewer #1: Yes

Reviewer #2: Yes

Reviewer #3: Partly

2. Has the statistical analysis been performed appropriately and rigorously? 

Reviewer #1: Yes

Reviewer #2: Yes

Reviewer #3: Yes

3. Have the authors made all data underlying the findings in their manuscript fully available?

Reviewer #1: Yes

Reviewer #2: Yes

Reviewer #3: No

4. Is the manuscript presented in an intelligible fashion and written in standard English?

Reviewer #1: Yes

Reviewer #2: Yes

Reviewer #3: No

5. Review Comments to the Author

Reviewer #1: The authors of this paper, whom I first congratulate for their incredible piece of research, identified the determinants of under-five mortality in Bangladesh by comparing Random Forest, Decision Tree, K-Nearest Neighbors, Logistic Regression, Support Vector Machine, XGBoost, LightGBM, and Neural Networks—effective machine learning techniques to identify these determinants. Random Forest was identified after a robust evaluation and was retained due to its high sensitivity and specificity, based on a higher area under the receiver operating characteristic curve.

Key determinants of under-five mortality in Bangladesh were found to include the number of household members, wealth index, parents’ education, the number of antenatal care (ANC) visits, birth order, and the father’s occupation. This study adds valuable insights into public health knowledge on the determinants of under-five mortality. In my opinion, their recommendations will not only improve policy but will also contribute to providing key data needed to develop longitudinal studies and intervention/implementation research aimed at improving the lives of under-five children in Bangladesh.

The paper is theoretically sound and professionally written. In my opinion, the authors present advanced machine learning models in a way that is accessible to peers and others who may not specialize in data science or machine learning. Reading this work was an incredible experience for me as I found it engaging and precise. The flow of writing keeps the reader captivated.

Overall, I consider the work to be theoretically sound, well-researched, well-written, interesting, and of significant public health importance.

Major Comments

1. Title and focus are not aligned

• The title suggests that the primary focus is on identifying the determinants of under-five mortality using machine learning algorithms. However, the reader is drawn into a methodological comparison of about seven advanced machine learning techniques, with little or no methodological backing on how these models were developed. Instead, a general overview of how these models were evaluated is provided.

• The objective emphasizes using advanced machine learning models to predict under-five mortality and its key determinants. However, the methodology involves comparing multiple machine learning models to determine the best-performing one before using that model to analyze the determinants.

• If the study’s main contribution is the comparison and selection of the best machine learning algorithm for predicting under-five mortality, this should be clearly reflected in the title. Moreover, identifying the determinants appears secondary in the current approach, as it follows the model evaluation process. While selecting the best model is a valuable task, it should not overshadow the importance of identifying determinants unless this is explicitly stated as the primary objective. If the primary goal is determinant identification, emphasize this in the methodology and results sections.

2. Insufficient methodological details

• The methods section is short and provides information about the model evaluation process and predictors, without insights into the model parameters for each machine learning algorithm. Additionally, details on how each machine learning model was developed, trained, and optimized are missing.

• This makes the methods section insufficient, as it lacks relevant information on how the models (Random Forest, Decision Tree, K-Nearest Neighbors, Logistic Regression, Support Vector Machine, XGBoost, LightGBM, and Neural Networks) were built, which would be of great interest to the reader.

• Other researchers should be able to replicate the study based on the methods provided. Without details on model parameters, this becomes impossible. It is also difficult to evaluate the fairness of the model comparisons without knowing the hyperparameters, evaluation metrics, and training/validation splits used.

3. Results section contains discussions

• The results section contains discussions of the findings rather than presenting only results. In my opinion, the authors should present just the results of their research in this section and reserve discussions for the discussion section.

Minor comments: No minor issues were identified

Reviewer #2: 1. Technical Soundness and Data Supporting Conclusions

Strengths:

• The study applies a range of machine learning (ML) algorithms, offering robust comparative analysis across models such as Random Forest, XGBoost, LightGBM, and others. The choice of algorithms aligns well with the complexity of the data.

• The Random Forest model's high predictive accuracy (e.g., 98.75% accuracy, 99% recall) strongly supports the study’s conclusions about key determinants of under-5 mortality.

• The use of the Boruta algorithm for feature selection ensures that identified predictors (e.g., household size, parental education, wealth index) are statistically significant and relevant.

Suggestions:

• Include a sensitivity analysis or additional validation (e.g., external validation) to further establish the robustness of the results.

• Provide a clearer acknowledgment of the limitations imposed by using cross-sectional data.

2. Statistical Analysis

Strengths:

• The study employs a comprehensive range of metrics (e.g., accuracy, precision, recall, F1 score, MCC) to evaluate model performance, enhancing the statistical rigor.

• K-fold cross-validation strengthens the reliability of the model comparisons..

Suggestions:

• Provide additional details about hyperparameter tuning, such as grid search ranges or optimization criteria.

• Clarify how the significance of variables in the chi-square tests translates into their importance in the ML models.

3. Data Availability

Strengths:

• The manuscript adheres to the PLOS ONE data policy, indicating that the BDHS 2022 dataset is publicly accessible through the Measure DHS website.

• Relevant data, such as variable definitions and coding, are included in the manuscript.

Suggestions:

• Ensure all relevant data, including feature importance scores and model configurations, are included in supplementary materials or a public repository.

• Revise the data availability statement to meet PLOS ONE's requirements explicitly.

4. Presentation and Language

Strengths:

• The manuscript is well-structured, with a logical flow from background to methods, results, and conclusions.

• Use of technical terms is appropriate, aiding clarity for a specialized audience.

Suggestions:

• Correct typographical errors and ensure consistency in tense throughout the manuscript.

• Strengthen the discussion by critically comparing the study’s findings with those of similar research, especially in the context of ML applications in health.

Reviewer #3: Congratulations on the work topic and the idea of investigating risk factors for mortality in children under 5 using multiple machine learning (ML) algorithms. However, some aspects of the article remain weak and need improvement.

GENERAL COMMENTS

1. The methodology needs to be better described, there is a lot of information missing, especially regarding the choice of sample and the data collected.

2. The results need to be reorganized, it is confusing to understand the main findings of the study.

3. The title, objectives, and conclusion need to be better aligned. Based on the title, one would expect a discussion of the main determinants of mortality, but the authors focus instead on comparing different machine learning (ML) algorithms. Additionally, the objectives are presented more than once throughout the text, creating redundancy, and the conclusion is overly long and does not adequately address the issue raised by the title.

SPECIFIC COMMENTS

The document does not specify line or page numbers for the review. For this reason, I have included some excerpts that I wanted to reference.

- Title: The title provides an acronym that is not familiar to most regions.

- Check keywords.

- “While traditional studies have been instrumental in identifying these risk factors, they often struggle to capture the complex, non-linear interactions between variables that contribute to under-five mortality”. Review expressions in English.

- “In doing so, this study has the potential to provide more nuanced insights into the factors driving under-five mortality in Bangladesh, making a substantial contribution to ongoing efforts to improve child survival rates and achieve the SDG targets.” – It seems like final considerations.

- Objectives are extensive. Determine the primary objective.

- What were the main determinants?

- What was the main model to predict mortality? This is only in the conclusion of the summary, but not in the conclusion of the article. Furthermore, this was not the objective of the work, and it was not suggestive in the title. Therefore, I suggest that authors adjust the title, objective and conclusion.

Methods

- Are the data from the entire population? Was the BDHS carried out throughout the country or just in one part?

- Sample: how many mothers there were in total. How many were excluded? I suggest you make a flowchart.

- “Temporary (de jure) residents...” Adjust expression.

Target variable

- The questions were from the BDHS survey or new data collection was carried out by the authors.

- There is no need to describe the encoding of the variables in the database, just explain that it was encoded in a binary variable (yes/no).

- How many children died? It is very important to report this number.

Independent variables

- Did this information also come from BDHS?

- It is important to briefly describe each variable. For example, “Maternal education level”, how many years each category has (0=No, 1=education, 2=Primary, 3=Secondary, 4=Higher), this may vary for each country.

- “Wealth index” – what makes up this variable? Was it linear and was it categorized?

- “Maternal age at first birth” so could it not correspond to the age when the child was born that the research mentioned?

- Have questions about prenatal care, and it was not clear on the methods of how the collection was carried out in relation to children. The mothers were asked if "Is your child alive?", but before that, were they asked if they had children? How many children? If the woman had more than one living child close in age, which of the children was chosen for inclusion of this prenatal data and in the study?

- Women who did not have children were excluded from the study, this is not described.

- Therefore, the mortality rate or total number of deaths was not used.

- This part of methods is very confusing, I suggest you review it so that the information is clear.

- “This study aims to investigate the risk factors associated with under-five mortality using various machine learning classification models” – Third objective? The objective is not stated here.

- Why was a multivariate analysis not performed?

Results

- Redundant, describe the % of those who died or those who did not. In addition to the percentage, include n (n=x).

- Why were children over 5 years old included if the analysis focused on deaths in children under 5? This was not addressed in the methods section.

- Table 1 is without legend and significant p values are indicated in bold, not symbols.

- Figures 1 and 2 use abbreviated variable names without description elsewhere in the text or in the caption.

- The first paragraph of results is very long.

- “In conclusion, the Random Forest consistently emerges as the best-performing model across all key metrics, including accuracy, precision, recall, AUC, MCC, Cohen's Kappa, and F1- score. It demonstrates strong predictive capabilities, outperforming other models slightly in almost every measure. LightGBM and XGBoost, while close competitors, perform similarly well in most metrics but fall just short of Random Forest's performance. Given its consistently high scores across all metrics, Random Forest is likely the best model overall, with LightGBM and XGBoost serving as excellent alternatives.” It's in the results section.

- Acronyms already described were described again throughout the work.

Discussion

- First paragraph: What were the main findings? Readjust paragraph.

- “The findings contribute valuable insights into understanding how various socio-economic, demographic, and health-related factors interact o influence child mortality” How do they contribute? This phrase seems to be out of place here, it resembles final considerations.

- “Family planning programs that provide access to contraceptives and reproductive health education could help reduce fertility rates, which, in turn, may lower under-5 mortality. Strengthening these programs in regions with high fertility and mortality rates should be prioritized.” Are the authors assuming that lowering fertility rates will reduce mortality rates? This assumption is problematic. Even a mother in a socially vulnerable situation who has only one child may face circumstances that increase the risk of infant death. Ensuring adequate access to basic sanitation, sufficient food, and proper prenatal care (including nutritional supplementation and an adequate number of visits), as well as addressing biological factors such as maternal age and exposure to pollutants, alcohol, and tobacco, are all crucial steps in reducing childhood mortality rates.

- “In conclusion, this study highlights the efficacy of machine learning in predicting determinants of under-5 mortality in Bangladesh. By leveraging the 2022 BDHS data, we were able to identify key socio-economic, demographic..." In the discussion and then there is another “conclusion” section.

Conclusion

- Much of the conclusion comprises final considerations that should be in the discussion. The conclusion, in general, should be concise and address the objective and title.

6. PLOS authors have the option to publish the peer review history of their article (what does this mean? ). If published, this will include your full peer review and any attached files.

**Do you want your identity to be public for this peer review?** For information about this choice, including consent withdrawal, please see our Privacy Policy .

Reviewer #1: **Yes: ** Fabrice Chethkwo

Reviewer #2: No

Reviewer #3: No

---

## [Author Response · Author response to Decision Letter 1]

4 Feb 2025

Dear Editors,

Regarding our manuscript, we thank you for the comments sent on December 20, 2024. My co-authors and I have revised the manuscript accordingly and would like it to be reconsidered for publication. As requested, we have included point-by-point responses to editors’ and reviewers’ comments below.

Please let us know if anything further is required at this time, and we thank you very much for considering our revised manuscript.

Shayla Naznin

---

## [Decision Letter · Decision Letter 1]

PONE-D-24-48508R1Predicting Determinants of Under-5 Mortality in Bangladesh Using Machine Learning Algorithm: Insights from BDHS 2022PLOS ONE

Dear Dr. Naznin,

Thank you for submitting your manuscript to PLOS ONE. After careful consideration, we feel that it has merit but does not fully meet PLOS ONE’s publication criteria as it currently stands. Therefore, we invite you to submit a revised version of the manuscript that addresses the points raised during the review process.

We look forward to receiving your revised manuscript.

Kind regards,

Md. Moyazzem Hossain, PhD

Academic Editor

PLOS ONE

Journal Requirements:

Reviewers' comments:

Reviewer's Responses to Questions

**Comments to the Author**

1. If the authors have adequately addressed your comments raised in a previous round of review and you feel that this manuscript is now acceptable for publication, you may indicate that here to bypass the “Comments to the Author” section, enter your conflict of interest statement in the “Confidential to Editor” section, and submit your "Accept" recommendation.

Reviewer #1: All comments have been addressed

Reviewer #2: All comments have been addressed

2. Is the manuscript technically sound, and do the data support the conclusions?

Reviewer #1: Yes

Reviewer #2: Yes

3. Has the statistical analysis been performed appropriately and rigorously? 

Reviewer #1: Yes

Reviewer #2: Yes

4. Have the authors made all data underlying the findings in their manuscript fully available?

Reviewer #1: Yes

Reviewer #2: Yes

5. Is the manuscript presented in an intelligible fashion and written in standard English?

Reviewer #1: Yes

Reviewer #2: Yes

6. Review Comments to the Author

Reviewer #1: The authors have revised the manuscript in response to my initial review, and all of my suggestions have been adequately addressed. It is of my opinion that the manuscript is suitable for publication.

Reviewer #2: Review Feedback

• Title and Abstract: The title is clear, and the abstract effectively summarizes the study's purpose, methodology, and significant findings. However, a more explicit statement about the implications of the findings could enhance clarity.

• Introduction: The introduction provides sufficient background on under-5 mortality and articulates the research objectives clearly. However, the significance of machine learning in this context could be emphasized more.

• Methods: The methods are described in detail, allowing for replication. Various machine learning models are explored; this comprehensive approach is appropriate for addressing the research question. A more detailed rationale for the choice of metrics for evaluation could enhance the methods section.

o Study Design: The study employs an appropriate design, leveraging machine learning models suited to the data type.

o Data Analysis: The statistical methods are appropriate and correctly applied. Reproducibility: The details provided allow for reproducibility

• Ethics: The stated compliance with ethical standards, including data source legality, is appropriate.

• Results: The results are presented clearly and supported by relevant data. Figures and tables are well-designed, aiding comprehension. Including confidence intervals alongside performance metrics could provide more context and robustness.

• Discussion: The discussion effectively interprets results within existing literature and acknowledges limitations, such as the cross-sectional nature of the data. A more detailed comparison with similar studies would strengthen the discussion.

• Conclusion: The conclusions drawn are supported by the data and effectively address the research objectives, reiterating key findings.

7. PLOS authors have the option to publish the peer review history of their article (what does this mean? ). If published, this will include your full peer review and any attached files.

**Do you want your identity to be public for this peer review?** For information about this choice, including consent withdrawal, please see our Privacy Policy .

Reviewer #1: No

Reviewer #2: No

---

## [Author Response · Author response to Decision Letter 2]

28 Apr 2025

I have revised my manuscript for your kind consideration.

---

## [Editor Report · Decision Letter 2]

Identifying Determinants of Under-5 Mortality in Bangladesh: A Machine Learning Approach with BDHS 2022 Data

PONE-D-24-48508R2

Dear Dr. Naznin,

We’re pleased to inform you that your manuscript has been judged scientifically suitable for publication and will be formally accepted for publication once it meets all outstanding technical requirements.

Kind regards,

Md. Moyazzem Hossain, PhD

Academic Editor

PLOS ONE
---

## [Editor Report · Acceptance letter]

PONE-D-24-48508R2

PLOS ONE

Dear Dr. Naznin,

I'm pleased to inform you that your manuscript has been deemed suitable for publication in PLOS ONE. Congratulations! Your manuscript is now being handed over to our production team.

Kind regards,

on behalf of

Professor Md. Moyazzem Hossain

Academic Editor

PLOS ONE